

# Psychometric properties of the Valuing Questionnaire in a Spaniard sample and factorial equivalence with a Colombian sample

Francisco J. Ruiz[1], Paula Odriozola-González[2], Juan C. Suárez-Falcón[3] and Miguel A. Segura-Vargas[1]

[1] Faculty of Psychology, Fundación Universitaria Konrad Lorenz, Bogotá, Colombia
[2] Department of Education, Universidad de Cantabria, Santander, Spain
[3] Faculty of Psychology, Universidad Nacional de Educación a Distancia, Madrid, Spain

## ABSTRACT

**Background**. The Valuing Questionnaire (VQ) is considered as one of the most psychometrically robust instruments to measure valued living according to the acceptance and commitment therapy model. It consists of 10 items that are responded to on a 7-point Likert-type scale and has two factors: Progression and Obstruction. The Spanish version of the VQ showed good psychometric properties in Colombian samples. However, there is no evidence of the psychometric properties of the VQ in Spaniard samples. This study aims to analyze the validity of the VQ in a large Spaniard sample and analyze the measurement invariance with a similar Colombian sample.

**Method**. The VQ was administered to a Spaniard sample of 846 adult participants from general online population. Cronbach's alpha and McDonald's omega were computed to analyze the internal consistency of the VQ. The fit of the VQ's two-factor model was tested through a confirmatory factor analysis with a robust maximum likelihood (MLR) estimation method. Afterward, we analyzed the measurement invariance across countries and gender. Convergent construct validity was analyzed with a package of questionnaires that evaluated experiential avoidance (Acceptance and Action Questionnaire-II, AAQ-II), emotional symptoms (Depression Anxiety and Stress Scale-21, DASS-21), life satisfaction (Satisfaction with Life Scale, SWLS), and cognitive fusion (Cognitive Fusion Questionnaire, CFQ).

**Results**. The internal consistency across samples was adequate (alphas and omegas were .85 for VQ-Progress and .84 for VQ-Obstruction). The two-factor model obtained a good fit to the data (RMSEA = 0.073, 90% CI [0.063, 0.083], CFI = 0.98, NNFI = 0.97, and SRMR = 0.053). The VQ showed strict invariance across countries and gender and showed theoretically coherent correlations with emotional symptoms, life satisfaction, experiential avoidance, and cognitive fusion. In conclusion, the Spanish version of the VQ demonstrated good psychometric properties in a large Spaniard sample.

Corresponding author
Francisco J. Ruiz,
franciscoj.ruizj@konradlorenz.edu.co

## INTRODUCTION

Acceptance and commitment therapy (ACT; *Hayes, Strosahl & Wilson, 1999*) is an empirically based psychological intervention (*Gloster et al., 2020*) that emphasizes the role of psychological flexibility on mental health and behavioral effectiveness. Psychological flexibility is usually defined in middle-level terms, which are higher-level functional abstractions that serve as shortcuts for applying basic principles to complex applied settings (*Vilardaga et al., 2009*). One of the most widely used definitions of psychological flexibility states that is "the ability to contact the present moment more fully as a conscious human being, and to change or persist in behavior when doing so serves valued ends" (*Hayes et al., 2006*, p. 7). As such, psychological flexibility comprises six core therapeutic processes: cognitive defusion (noticing thinking in flight in a detached and non-judgmental way), acceptance (being open to experience unpleasant private experiences), contact with the present moment (attention flexibility), self-as-context (noticing ongoing behavior from an inclusive and transcendent perspective), values (verbally constructed positive reinforcers), and committed action (build and broaden values-based behavioral patterns). Values is a crucial process in the psychological flexibility model because they provide direction and meaning to the individual's behavior.

More specifically, values are conceptualized as verbally constructed positive reinforcers that are at the top of a hierarchy of reinforcers, including goals and more tangible reinforcers (*Barnes-Holmes et al., 2004*; *Gil-Luciano et al., 2019*; *Luciano, Valdivia-Salas & Ruiz, 2012*). Values entail intrinsically reinforced dynamic patterns of activity that lead the individual symbolically closer to his/her values *Wilson & DuFrene, 2009*). The specific activities that would permit advancing towards own values frequently vary over time and across situations. For instance, a valued action for an undergraduate whose exams are getting close would probably be studying, whereas on holidays, valued actions might be related to activities involving social relationships.

According to the ACT model, values are related to joy, meaning, and suffering in two main ways (*Ruiz, 2020*). Firstly, thoughts and actions symbolically related to these hierarchical positive reinforcers acquire intense appetitive functions (*Gil et al., 2012*; *Gil-Luciano et al., 2019*). For instance, when valuing a romantic relationship characterized by sharing and transparency, telling a traumatic experience will acquire appetitive functions that might undermine the aversive functions actualized when sharing that experience. Similarly, thoughts and actions symbolically related in opposition to these hierarchical positive reinforcers (*i.e.,* values) will acquire intense aversive functions (*Gil-Luciano et al., 2019*; *Ruiz et al., 2020a*). In summary, values and suffering usually become the two sides of the same coin (*Wilson & DuFrene, 2009*). Secondly, suffering is exacerbated as a consequence of displaying an inflexible pattern of behavior characterized by the entanglement with unpleasant private experiences and engagement in experiential avoidance strategies (*Hayes et al., 2006*). This behavioral inflexibility reduces the frequency of valued actions, leading to experiencing more aversive thoughts and emotions in opposition to values (*Ruiz et al., 2020c*).

Given the central role of values in the ACT model, numerous self-report measures of this process have been developed in the last decade, and three recent systematic reviews have been published (*Barrett, O'Connor & McHugh, 2019*; *Reilly et al., 2019*; *Serowik et al., 2018*). Some of the most used instruments of values are the Valued Living Questionnaire (VLQ; *Wilson et al., 2010*), the Bull's-Eye Values Survey (BEVS; *Lundgren et al., 2012*), the Engaged Living Scale (ELS; *Trompetter et al., 2013*), and the Valuing Questionnaire (VQ; *Smout et al., 2014*). Values self-reports can be grouped according to their procedure. Some questionnaires rate to what extent participants value pre-established different life domains and the consistency of their behavior according to their values (*e.g.*, VLQ, BVES). Other instruments measure overall valued living without specifying pre-established life domains (ELS and VQ). Values instruments that explore life domains are more informative than general measures, but they have the limitation that are time-consuming. General values measures are often short and easier to score and interpret.

According to the systematic reviews mentioned before, values instruments vary in their psychometric quality. Two of these reviews indicate that the VQ is probably the most psychometrically robust instrument (*Barrett, O'Connor & McHugh, 2019*; *Reilly et al., 2019*). In the original study of the VQ, *Smout et al. (2014)* asked 630 undergraduates to grade a pool of 70 items, which representativeness was previously rated by eight ACT experts. By using a Rating Scale Model and a Partial Credit Model, the authors retained 10 of the 70 original items. These 10 items were administered to a second sample. The final version of the VQ showed a two-factor structure with the following subscales: Progress (*i.e.,* enactment of values, including clear awareness of what is personally meaningful and perseverance) and Obstruction (*i.e.,* disruption of valued living due to avoidance of unwanted experience and distraction from values). The internal consistency of both subscales was good (from .81 to .87 for Progress and from .79 to .87 for Obstruction). The VQ-Progress scores showed strong correlations with positive affect, well-being, life satisfaction, and mindfulness skills. Conversely, scores on the VQ-Obstruction showed strong positive correlations with emotional symptoms, negative affect, and experiential avoidance.

Across studies, the VQ has shown excellent psychometric properties, with good internal consistency, structural validity, and a higher sensitivity to treatment than other values measures (*Barrett, O'Connor & McHugh, 2019*; *Reilly et al., 2019*). The convergent construct validity of the VQ has been supported in subsequent studies (*Carvalho et al., 2018*; *Rickardsson et al., 2019*; *Ruiz et al., 2021*).

Unfortunately, the psychometric properties of the VQ have not been widely studied across different cultures and countries. To our best knowledge, there are only three validation studies of the VQ in other languages. Two studies analyzed the validity of the VQ in the context of chronic pain in Portuguese and Swedish (*Carvalho et al., 2018*; *Rickardsson et al., 2019*). In the remaining study, *Ruiz et al. (2021)* administered the Spanish version of the VQ to three Colombian samples. The VQ showed good internal consistency across samples, and the two-factor model obtained a good fit to the data. Additionally, scalar measurement invariance was found across clinical and nonclinical samples and gender.

To our best knowledge, no studies have been conducted in Spain regarding the psychometric properties of the VQ. This could hinder research aimed to study the process of values from the ACT standpoint with Spaniard samples. Additionally, no studies have explored the measurement invariance across cultures. Therefore, this study aimed to analyze the validity of the VQ with a Spaniard sample and the factorial equivalence with a Colombian sample. We expected that the VQ would show good internal consistency and a good fit of the two-factor structure in the Spaniard sample. Additionally, we expected to find measurement invariance across countries and gender. Likewise, we expected to find positive correlations between VQ-Progress and life satisfaction and negative correlations with emotional symptoms, experiential avoidance, and cognitive fusion. An inverse pattern of correlations was expected between VQ-Obstruction and the remaining constructs.

## MATERIALS & METHODS

According to *Ato, López-García & Benavente (2013)*, an instrumental design was implemented by analyzing the psychometric properties of the VQ and its measurement invariance across gender and countries. The procedures implemented in this research were approved by the Bioethics Committee of Fundación Universitaria Konrad Lorenz (2016-021B). Informed written consents were obtained from all participants in this study.

### Procedure

Participants in both samples responded to an anonymous online survey distributed through social media (*e.g.*, institutional webpages, Facebook, etc.) through a snowball sampling procedure. Specifically, the researchers asked their contacts and participants to share the publication of the survey to reach more people. The research publication was not posted in specific groups (*e.g.*, students' or mental health groups).

Participants first provided informed consent by accepting the conditions described at the beginning of the survey that included that they were adults (*i.e.,* the only exclusion criterion to participate being younger than 18 years). Afterward, the participants in Sample 1 completed the instruments presented above. Participants in Sample 2 responded to a similar survey, but only the VQ data are relevant for this study. Their respective instructions preceded each questionnaire. Participants were also emphasized that the survey was anonymous and that they could stop their participation anytime they wanted. The median time for the completion of surveys was approximately 15 min. When data collection was finished, global reports of the results of the surveys were sent to the participants who indicated that they were interested in them. Participants were not compensated for their participation.

The survey conducted to recruit Sample 1 was available for one year until at least 200 participants of both genders responded. As we planned to analyze measurement invariance across gender, this number was established because it is the minimum suggested, as a rule of thumb, for conducting confirmatory factor analyses (*Kline, 2016*).

### Participants

*Sample 1*. This sample consisted of 846 Spanish participants (75.7% females) with an age range between 18 and 72 ($M = 35.40$, $SD = 11.39$). The participants' relative education

level was 0.1% no studies, 33% primary studies (*i.e.,* compulsory education) or mid-level study graduates (*i.e.,* high school or vocational training), and 65.6% were undergraduates or college graduates (1.3% did not indicate the educational level). Almost half of the participants (44.6%) reported having received psychological or psychiatric treatment in the past, but only 12.8% were currently in treatment. Also, 12.9% of participants reported using psychotropic medication.

*Sample 2.* This sample consisted of 724 Colombian participants (74.4% females) whose ages ranged between 18 and 88 ($M = 26.11$, $SD = 8.93$). The participants' relative education level was 17.8% primary studies or mid-level study, and 82.2% were undergraduates or college graduates. Forty-five percent reported having received psychological or psychiatric treatment at some time, but only 8.4% were currently in treatment. Also, 5.4% of participants reported using psychotropic medication.

**Instruments**

*Valuing Questionnaire* (VQ; *Smout et al., 2014*; Spanish translation by *Ruiz et al., 2021*). The VQ is a 10-item, 7-point Likert (6 = *completely true*; 0 = *not at all true*), self-report instrument designed to assess general valued living over the last week. It assesses valued living in everyday language without referring to specific life domains. The VQ has two 5-item factors: Progress (*i.e.,* enactment of values, including clear awareness of what is personally meaningful and perseverance) and Obstruction (*i.e.,* disruption of valued living due to avoidance of unwanted experiences and distraction from values). The Spanish version of the VQ has shown good psychometric properties in Colombian samples, with alphas of .83 and .82 for Progress and Obstruction, respectively.

*Acceptance and Action Questionnaire –II* (AAQ-II; *Bond et al., 2011*; Spanish version by *Ruiz et al., 2013*). The AAQ-II consists of 7 items that measure experiential avoidance on a 7-point Likert-type scale (7 = *always true*; 1 = *never true*). The Spanish version of the AAQ-II has shown good psychometric properties (mean alpha of .88) and a one-factor structure in Spanish samples (*Ruiz et al., 2013*). In this study, the AAQ-II showed an alpha of .91. Strong positive correlations were expected between AAQ-II and VQ-Obstruction, whereas at least medium negative correlations were expected with VQ-Progress.

*Cognitive Fusion Questionnaire* (*Gillanders et al., 2014*; Spanish version by *Ruiz et al. (2017a)*. The CFQ is a 7-item, 7-point Likert-type scale (7 = *always true*; 1 = *never true*) of general cognitive fusion. The Spanish version by *Ruiz et al. (2017a)* has shown excellent internal consistency (alpha of .92) and a one-factor structure. In this study, the AAQ-II showed an alpha of .93. As with the AAQ-II, strong positive correlations were expected between CFQ and VQ-Obstruction, whereas at least medium negative correlations were expected with VQ-Progress.

*Depression, Anxiety, and Stress Scales –21* (DASS-21; *Lovibond & Lovibond, 1995*; Spanish version by *Daza et al., 2002*). The DASS-21 is a 21-item, 4-point Likert-type scale (3 = *applied to me very much, or most of the time*; 0 = *did not apply to me at all*) consisting of sentences describing negative emotional states: Depression, Anxiety, and Stress. The Spanish version of the DASS-21 has shown good internal consistency for all the subscales and a hierarchical factor structure with a second-order factor (*Ruiz et al., 2017b*).

In this study, the DASS-21 obtained alphas of .95, .92, .87, and .86 for the total scale, Depression, Anxiety, and Stress, respectively. Strong positive correlations were expected between the DASS-21 and VQ-Obstruction, whereas at least medium negative correlations were expected with VQ-Progress.

*Satisfaction with Life Survey* (SWLS; *Diener et al., 1985*; Spanish translation by *Atienza et al., 2000*). The SWLS is a 5-item survey that evaluates self-perceived well-being through a 7-point Likert-type scale (7 = *strongly agree*; 1 = *strongly disagree*). *Ruiz et al. (2019)* found that the Spanish version of the SWLS showed good psychometric properties and convergent validity in a Spaniard sample. In this study, the SWLS showed an alpha of .89. Strong positive correlations were expected between the SWLS and VQ-Progress, and medium negative correlations were expected with VQ-Obstruction.

### Statistical and psychometric analysis

Firstly, we analyzed the fit of the two-factor model of the VQ by computing confirmatory factor analyses (CFAs) with Sample 1 (*i.e.,* the Spaniard sample). In so doing, we followed the procedure used in the Spanish validation of the VQ in Colombia (*Ruiz et al., 2021*). As previous CFAs of the VQ have shown a method effect in Items 5 and 7 responses (*Smout et al., 2014*), we decided to compare the fit of the two-factor model where the error terms between these items were allowed to correlate *versus* the two-factor model with no error correlations. The software LISREL© (version 8.71, *Jöreskog & Sörbom, 1999*) was used to conduct the CFAs. No missing values were found due to the collection method (*i.e.,* online survey). Given the lack of multivariate normality in the data (multivariate Mardias' test of skewness and kurtosis = 610.954; $p < .001$), we selected the robust maximum likelihood (MLR) estimation method with the covariance matrix and the asymptotic variance–covariance matrix. The robust estimation methods in structural equation modeling are known for overcoming to a great extent the problem of the presence of outliers (*Yuan & Zong, 2013*). Even so, we have also used z-scores and Mahalanobis distance and obtained virtually the same results with or without the few identified outliers.

We computed the Satorra-Bentler chi-square test and the following goodness-of-fit indexes for the two-factor model: (a) the comparative fit index (CFI), (b) the non-normed fit index (NNFI), (c) the root mean square error of approximation (RMSEA), (d) the parsimony normed fit index (PNFI), and (e) the standardized root mean square residual (SRMR). According to authoritative guidelines (*e.g., Hu & Bentler, 1999*), CFI and NNFI values above .90 represent acceptable models, and above .95 indicate a good fit to the data. Regarding the RMSEA, values below 0.08 represent an acceptable fit, and values below 0.05 constitute a good fit (*Browne & Cudeck, 1992*; *Browne & Cudeck, 1993*; *Hooper, Coughlan, & Mullen, 2008*). Concerning the SRMR, values of 0.08 represent a good fit, and values below 0.05 represent a very good fit to the data. Furthermore, higher PNFI values indicate a more parsimonious model. Lastly, we also calculated the *p*-value for test of close fit (PCLOSE), which computes a one-sided test of the null hypothesis that RMSEA is equal or lower than 0.05.

Secondly, we analyzed construct reliability and evidence of convergent and discriminant validity of the measurement model following the suggestions by *Brown (2015)* and *Fornell*

& *Larcker (1981)*. The composite reliability coefficient (CR) was computed to analyze construct reliability. CR values higher than 0.70 can be considered as high construct reliability and adequate internal consistency. The convergent validity of the measurement model was analyzed according to three criteria: (a) factor loadings should be statistically significant (standardized loadings estimates should be 0.40 or higher), (b) CR should be higher than 0.70, and (c) the average variance extracted (AVE) should be $\geq 0.50$ for each subscale of the VQ. Lastly, discriminant validity was estimated according to the following criteria: (a) inter-construct correlations should be lower than 0.80, and (b) the square root of AVE ($\sqrt{AVE}$) of both factors of the VQ should be greater than the inter-construct correlations with any other factor.

Thirdly, we analyzed the measurement invariance across countries (*i.e.,* Spain and Colombia) and gender. In so doing, we followed also the procedure used by *Ruiz et al. (2021)* in the Spanish validation of the VQ in Colombia. Specifically, we adopted the guidelines proposed by *Jöreskog (2005)* and *Millsap & Yun-Tein (2004)*. Through additional CFAs, we tested metric, scalar, and strict invariances by analyzing whether the item factor loadings, items intercepts, and the variance of error of the items were invariant across countries and gender. The relative fit of four progressively more restrictive models was compared. Firstly, the multiple-group baseline model allowed the unstandardized factor loadings to vary across groups. Conversely, the pattern of item-factor loadings and the number of factors were the same across groups (configural invariance). Secondly, the metric invariance model was nested within the previous model and placed equality of factor loadings across groups (*i.e.,* weak factorial invariance). Thirdly, the scalar invariance model was nested within the metric invariance model and confined the factor loadings and the items intercepts to be the same across groups (*i.e.,* strong factorial invariance). Lastly, the strict invariance model was nested within the scalar invariance and assumed the variance of errors (*i.e.,* indicator residuals) to be equal across groups. Following *Kline (2005)* indications, we did not place equality constraints on estimates of the factor variances because they differ across groups even when the indicators are quantifying the same construct in an equivalent manner. For the model comparison, we weighed the CFI, NNFI, and RMSEA indices between nested models. We chose the more constrained model (*i.e.,* second model *versus* the first model, and third model *versus* the second model) following the criteria advocated by *Cheung & Rensvold (2002)* and *Chen (2007)*: (a) the difference in RMSEA ($\Delta$RMSEA) was lower than .01; (b) the differences in NNFI ($\Delta$NNFI) and CFI ($\Delta$CFI) were equal to or higher than $-.01$.

Fourthly, we analyzed the internal consistency of the VQ in Sample 1 with the MBESS package in R (*Kelley & Lai, 2012*; *Kelley & Pornprasertmanit, 2016*). We computed corrected item-total correlations to identify items that should be removed because of showing a low discrimination item index (*i.e.,* values below .30). Then, we calculated coefficient alphas and McDonald's omegas and provided percentile bootstrap 95% confidence intervals (CI). According to *Nunnally & Bernstein (1994)*, values higher than .70 were considered acceptable, whereas above .80 were considered good. To determine what label would apply to the VQ subscales, we observed the 95% CI. Fifthly, we calculated Pearson correlations between the VQ and the remaining scales to assess evidence based on relationships with
Table 1 Goodness-of-fit indexes of the two-factor model and the two-factor model with error terms allowed to correlate for items 5 and 7 in sample 1.

| Goodness-of-fit indicators | Two factor-model | Two factor-model with error terms allowed to correlate (items 5 and 7) |
|---|---|---|
| RMSEA [90% CI] | 0.073[0.063, 0.083] | 0.073 [0.063, 0.084] |
| CFI | 0.98 | 0.98 |
| NNFI | 0.97 | 0.97 |
| SRMR | 0.053 | 0.052 |
| PNFI | 0.74 | 0.71 |
| $S\text{-}B\chi^2$ $(df)$ | 185.790 (34) | 182.686 (33) |

Notes.
CFI, Comparative Fit Index; VQ, Valuing Questionnaire; NNFI, Non-Normed Fit Index; RMSEA, Root Mean Square Error of Approximation; $S\text{-}B\chi^2$, Satorra-Bentler Chi-Square Test; SRMR, Standardized Root Mean Square Residual.

other variables using SPSS 25©. Correlations were interpreted according to the guidelines provided by *Lenhard & Lenhard (2016)*: small correlation between .10 and .20, medium between .21 and .36, and strong correlations >.36. Lastly, we calculated descriptive data and explored differences in VQ scores across country and gender by computing independent sample t-tests.

# RESULTS

## Validity evidence based on internal structure
### Dimensionality
Table 1 shows the results of the CFAs conducted in Sample 1. The fit of the two-factor model of the VQ was good according to the values of CFI (.98), NNFI (.97), and SRMR (0.053). The RMSEA value was 0.073, which indicates an acceptable fit. However, the test of close fit (PCLOSE) showed that the RMSEA value was significantly higher than 0.05 ($p < .01$). Thus, overall, the two-factor model showed an acceptable fit according to this goodness-of-fit index. The alternative two-factor model with correlated error terms in Items 5 and 7 did not significantly improve the fit. In conclusion, we chose the two-factor model due to its greater parsimony according to the PNFI (see Table 1). This was also the factor model selected in *Ruiz et al. (2021)* with Colombian participants. Figure 1 depicts the results of the completely standardized solution of the two-factor model in Sample 1.

### Construct reliability and convergent and discriminant validity of the measurement model
The VQ showed high construct reliability because the CR values were higher than 0.70 for both VQ-Progression (0.85) and VQ-Obstruction (0.84). The convergent validity of the VQ was also supported. First, all factor loadings were statistically significant and higher than 0.40 (with six items exceeding the ideal cutoff of 0.70). Second, the AVE values of both VQ-Progress (0.54) and VQ-Obstruction (0.52) were higher than 0.50. Lastly, the CR values of both constructs were very high. The VQ also obtained indicators of appropriate discriminant validity. First, the inter-construct correlation was −0.60, which is lower in absolute value than 0.80. Second, the square roots of AVE were greater than

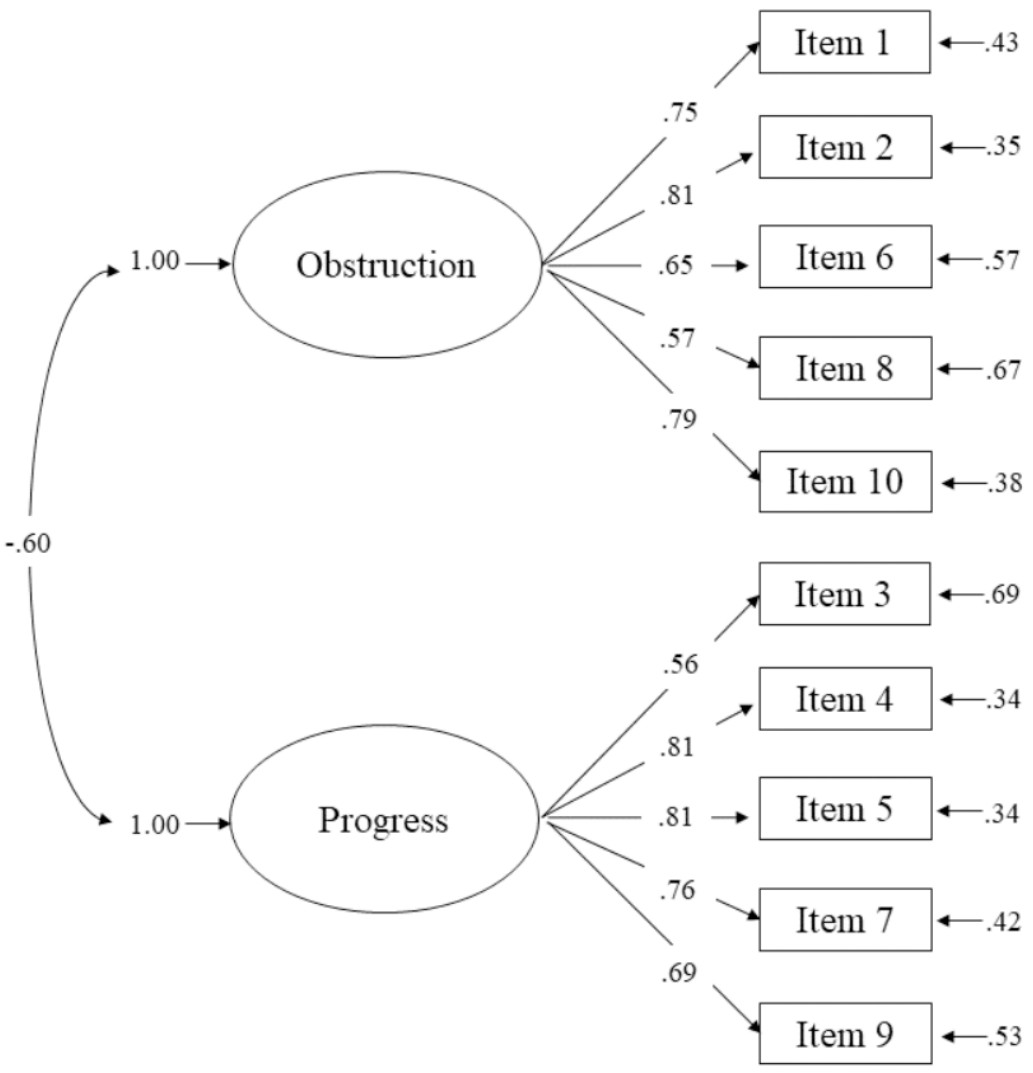

**Figure 1   Completely standardized solution for the two-factor model of the Valuing Questionnaire in Sample 1.**

the inter-construct correlation of this latent variable with any of the factors (VQ-Progress: $\sqrt{AVE} = 0.73$; VQ-Obstruction = ($\sqrt{AVE} = 0.72$).

### Measurement invariance

Table 2 presents the results of the metric, scalar, and strict invariance analyses. Parameter invariance was supported at all levels across countries (Spain and Colombia) and gender because there were no differences higher than 0.01 in the RMSEA, CFI, and NNFI favoring the models with lower constraints. Thus, measurement invariance across countries and gender was demonstrated.
**Table 2   Results of the metric and scalar invariance analyses across Spain and Colombia and gender.**

| Model | RMSEA | ΔRMSEA | CFI | ΔCFI | NNFI | ΔNNFI |
|---|---|---|---|---|---|---|
| *Measurement invariance across countries* | | | | | | |
| MG Baseline model | 0.075 | | 0.976 | | 0.969 | |
| Metric invariance | 0.071 | 0.004 | 0.976 | 0.000 | 0.972 | 0.003 |
| Scalar invariance | 0.070 | 0.001 | 0.974 | −0.002 | 0.972 | 0.000 |
| Strict invariance | 0.065 | 0.005 | 0.975 | 0.001 | 0.976 | 0.004 |
| *Measurement invariance across gender* | | | | | | |
| MG Baseline model | 0.075 | | 0.976 | | 0.969 | |
| Metric invariance | 0.071 | 0.004 | 0.976 | 0.000 | 0.972 | 0.003 |
| Scalar invariance | 0.069 | 0.002 | 0.975 | −0.001 | 0.973 | 0.001 |
| Strict invariance | 0.065 | 0.004 | 0.975 | 0.000 | 0.976 | 0.003 |

## Psychometric quality of the items

Table 3 shows that all corrected item-total correlations of the VQ in Sample 1 were high. For VQ-Progress, they ranged from .52 to .71 in Sample 1, whereas for VQ-Obstruction from .51 to .70. Cronbach's alpha coefficients were adequate for both subscales (VQ-Progress = .85, 95% CI [.83–86]; VQ-Obstruction = .84, 95% CI [.82–.86]). The values of McDonald's omega coefficients were virtually the same (VQ-Progress = .85, 95% CI [.83–87]; VQ-Obstruction = .84, 95% CI [.82–.86]).

## Validity evidence based on relationships with other variables

Table 4 presents the Pearson correlations between the VQ and the assessed constructs. Overall, the VQ showed correlations in the expected directions and size. Specifically, VQ-Progress showed strong negative correlations with experiential avoidance and cognitive fusion. Medium-size correlations were found between VQ-Progress and emotional symptoms measured with the DASS-21. Lastly, VQ-Progress correlated positively and strongly with life satisfaction. As expected, VQ-Obstruction obtained the opposite pattern of correlations. Specifically, VQ-Obstruction showed strong and positive correlations with experiential avoidance, cognitive fusion, and emotional symptoms. VQ-Obstruction correlated negatively and strongly with life satisfaction.

## Scores on the VQ across countries and gender

No statistically significant differences were found on VQ scores across countries on VQ-Progress (Spain: $M = 19.08$, $SD = 6.08$; Colombia: $M = 19.50$, $SD = 6.43$; $t(1568) = -1.35$, $p = .18$) and VQ-Obstruction (Spain: $M = 11.31$, $SD = 6.63$; Colombia: $M = 11.70$, $SD = 6.88$; $t(1568) = -1.16$, $p = .25$). Regarding gender in Spaniard participants (*i.e.*, Sample 1), there were no statistically significant differences neither for VQ-Progress (Men: $M = 18.72$, $SD = 6.13$; Women: $M = 19.14$, $SD = 6.08$; $t(811) = -0.80$, $p = .43$) or VQ-Obstruction (Men: $M = 11.14$, $SD = 6.60$; Women: $M = 11.39$, $SD = 6.69$; $t(811) = -0.43$, $p = .67$).

**Table 3   Item description and corrected item-total correlations in sample 1.**

| Item number and description | Corrected item-total correlations |
|---|---|
| 1. Pasé un montón de tiempo pensando sobre el pasado o el futuro en vez de dedicarme a actividades que eran importantes para mí [I spent a lot of time thinking about the past or future, rather than being engaged in activities that mattered to me]. OBSTRUCTION | .69 |
| 2. Estuve básicamente en "piloto automático" la mayor parte del tiempo [I was basically on "auto-pilot" most of the time]. OBSTRUCTION | .70 |
| 3. Trabajé para conseguir mis metas incluso cuando no me sentía motivado [I worked toward my goals even if I didn't feel motivated to]. PROGRESS | .52 |
| 4. Estuve orgulloso de cómo viví mi vida [I was proud about how I lived my life]. PROGRESS | .71 |
| 5. Hice progresos en las áreas de mi vida que más me importan [I made progress in the areas of my life I care most about]. PROGRESS | .71 |
| 6. Los pensamientos, sentimientos y recuerdos difíciles se interpusieron en el camino de lo que quería hacer [Difficult thoughts, feelings or memories got in the way of what I really wanted to do]. OBSTRUCTION | .62 |
| 7. Continué mejorando en ser el tipo de persona que deseo ser [I continued to get better at being the kind of person I want to be]. PROGRESS | .71 |
| 8. Cuando las cosas no fueron según lo planeado, me di por vencido fácilmente [When things didn't go according to plan, I gave up easily]. OBSTRUCTION | .51 |
| 9. Me sentí como si tuviera un propósito en la vida [I felt like I had a purpose in life]. PROGRESS | .62 |
| 10. Parecía como si estuviera comportándome de manera mecánica en vez de centrarme en lo que era importante para mí [It seemed like I was just "going through the motions" rather than focusing on what was important to me]. OBSTRUCTION | .69 |

# DISCUSSION

The VQ is one of the most widely used and psychometrically robust measures of valued living according to the ACT model (*e.g.*, *Barrett, O'Connor & McHugh, 2019*; *Reilly et al., 2019*; *Reilly et al., 2019*; *Serowik et al., 2018*). Although there is a Spanish version of the VQ (*Ruiz et al., 2021*), no studies have analyzed its psychometric properties in Spaniard samples. The current study aimed to fill this gap by analyzing the psychometric properties and factor structure of the Spanish version of the VQ with a large Spaniard sample ($N = 846$). Additionally, we analyzed the factorial equivalence of the VQ between the Spaniard sample and a Colombian sample ($N = 724$) with similar characteristics.

The results showed that the VQ obtained good psychometric properties in the Spaniard sample. Regarding internal consistency, the VQ showed appropriate Cronbach's alpha and

**Table 4** Pearson correlations between the VQ scores and other relevant self-report measures in sample 1.

| Measure | r with progress | r with obstruction |
|---|---|---|
| VQ-Obstruction | −.51[*] | |
| AAQ-II (Experiential avoidance) | −.49[*] | .66[*] |
| CFQ (Cognitive fusion) | −.44[*] | .68[*] |
| DASS-Total | −.36[*] | .65[*] |
| DASS-21 –Depression | −.48[*] | .66[*] |
| DASS-21 –Anxiety | −.23[*] | .53[*] |
| DASS-21 –Stress | −.24[*] | .54[*] |
| SWLS (Life satisfaction) | .64[*] | −.53[*] |

Notes.

AAQ-II, Acceptance and Action Questionnaire –II; DASS - 21, Depression, Anxiety, and Stress Scales –21; SWLS, Satisfaction with Life Scale; VQ, Valuing Questionnaire; SWLS, Satisfaction with Life Scale.

[*] $p < .001$.

McDonald's omega values for both Progress (.85) and Obstruction (.84). These values are in the range of previous validation studies (*e.g.*, *Barrett, O'Connor & McHugh, 2019*; *Ruiz et al., 2021*; *Smout et al., 2014*). The two-factor model of the VQ obtained a good fit to the data. The fit of this model was very similar to the two-factor model with correlated error terms in Items 5 and 7, which was the factor model considered as more appropriate in the original validation study (*Smout et al., 2014*). As in *Ruiz et al. (2021)*, we chose the two-factor model due to its greater parsimony. Additionally, the measurement model of the VQ showed adequate construct reliability and convergent and discriminant validity.

The VQ also showed measurement invariance at metric, scalar, and strict levels across countries and gender. The correlations of the VQ with other related instruments were theoretically coherent and equivalent to those found in other studies (*Ruiz et al., 2021*; *Smout et al., 2014*). Specifically, VQ-Progress was positively correlated with life satisfaction and negatively with emotional symptoms, experiential avoidance, and cognitive fusion. As expected, the opposite pattern of correlations was found between VQ-Obstruction and the constructs mentioned above. Thus, these results add further empirical evidence of the adaptive role of valued living according to the ACT model.

The Spanish version of the VQ showed similar psychometric properties in this Spaniard sample to the previous study with Colombian samples (*Ruiz et al., 2021*). Accordingly, this study presents further evidence of the robust psychometric properties of the VQ (*e.g.*, *Barrett, O'Connor & McHugh, 2019*; *Reilly et al., 2019*). To our best knowledge, this study presents the first analysis of factorial equivalence of the VQ across different cultures. The evidence of scalar invariance is relevant because it permits comparing scores across Spaniard and Colombian samples (*Greiff & Scherer, 2018*). In this regard, the Spaniard and Colombian samples analyzed in this study showed similar mean scores on the VQ subscales, which preliminarily indicates that there seems to be similar levels of valued living in these countries. Further studies might compare valued living scores across larger Spaniard and Colombian samples This study also adds evidence of the factorial equivalence of the Spanish version of the VQ across gender. As in *Ruiz et al. (2021)*, the VQ mean scores

did not differ across gender. Thus, it seems that valued living does not differ across gender in these two Spanish-speaking countries.

It is worth mentioning some of the limitations of this study. Firstly, the VQ was not administered to a clinical sample. Further studies should analyze the psychometric properties of the VQ in Spaniard clinical participants and the measurement invariance across clinical and nonclinical individuals. Given that the study conducted by *Ruiz et al. (2021)* found strict measurement invariance of the VQ across Colombian clinical and nonclinical participants, we would expect to find similar results in Spaniard samples. Secondly, the percentage of women was significantly higher than the percentage of men in the Spaniard sample. However, this limitation was reduced by the results obtained in the measurement invariance across gender. Thirdly, the VQ was only correlated with other self-reports, which could have inflated the correlations found among the VQ subscales and the other instruments. Fourthly, the participants' mean age was relatively low, which might be a consequence of the more frequent use of the internet and social media by young people in Spain. Further studies should explore the psychometric properties of the VQ in samples of older participants. Lastly, we did not test the treatment sensitivity of the VQ in Spaniard samples. Subsequent studies should analyze this issue. It would be expected that the VQ would show treatment sensitivity in Spaniard samples according to previous research that used the Spanish version of the VQ in Colombia (*e.g.*, *Ruiz et al., 2018*; *Ruiz et al., 2020b*).

Despite these limitations, the current study has some practical implications for Spaniard researchers and mental health professionals. Firstly, the VQ instrument might be used in several research contexts, such as in survey studies that analyze the role of values in mental health and clinical studies analyzing the efficacy and processes of change in ACT interventions. Secondly, the VQ can be adopted in the routine assessment conducted by ACT practitioners. Lastly, the VQ can be used by researchers that aim to compare valued living across gender and Spain and Colombia.

## CONCLUSIONS

The findings of this study are consistent with the previous analysis of the psychometric properties of the Spanish version of the VQ in Colombian samples (*Ruiz et al., 2021*). The current study adds empirical evidence of the good fit of the two-factor model of the VQ and its measurement invariance across Spaniard and Colombian samples. The factorial equivalence of the Spanish version of the VQ permitted comparing its scores across these countries. Further studies might analyze the measurement invariance of the VQ across additional Spanish-speaking countries and across different languages.

In conclusion, the VQ showed to be a reliable and valid instrument to measure valued living according to the ACT model in a large Spaniard sample.

### Funding

The authors received no funding for this work.

### Competing Interests

The authors declare there are no competing interests.

### Author Contributions

- Francisco J. Ruiz conceived and designed the experiments, performed the experiments, analyzed the data, prepared figures and/or tables, authored or reviewed drafts of the paper, and approved the final draft.
- Paula Odriozola-González conceived and designed the experiments, performed the experiments, authored or reviewed drafts of the paper, and approved the final draft.
- Juan C. Suárez-Falcón conceived and designed the experiments, analyzed the data, prepared figures and/or tables, authored or reviewed drafts of the paper, and approved the final draft.
- Miguel A. Segura-Vargas performed the experiments, prepared figures and/or tables, authored or reviewed drafts of the paper, and approved the final draft.

### Human Ethics

The following information was supplied relating to ethical approvals (i.e., approving body and any reference numbers):

The procedures followed in the research reported in the manuscript were approved by the Bioethics Committee of Fundación Universitaria Konrad Lorenz (2016-021B).

### Data Availability

All responses to the VQ items, the gender of the participants, and the sample they pertained to are available in the Supplemental File.

### Supplemental Information

Supplemental information for this article can be found online at http://dx.doi.org/10.7717/peerj.12670#supplemental-information.

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
