# Peer review of "Psychometric properties of the Valuing Questionnaire in a Spaniard sample and factorial equivalence with a Colombian sample"

_PeerJ, doi:10.7717/peerj.12670_

## Round 0.1 · original submission · Major Revisions

Dear authors , please find enclosed the comments provided by the school reviewers . Please kindly make diligent effort to attend to all of them, providing all the necessary details. Your work will definitely improve. Looking forward to your revised manuscript.
Thank you very much

Reviewer 1 ·

Basic reporting

no comment

Experimental design

Material & Methods
3. I think the description of the procedures (lines 191 to 199) could be presented before the participants (line 139).
4. What was the method used to estimate the sample size of this study?
5. What are the inclusion and exclusion criteria of this study? How many participants accepted participated in this study?

Instruments
6. Can the authors add the VQ’s alpha of the original study, as they do with the other scales?

Validity of the findings

Discussion
The authors could discuss the practical implications of the current study.

Additional comments

Psychometric properties of the Valuing Questionnaire in a Spaniard sample and factorial equivalence with a Colombian sample

I am very grateful for the opportunity to review this manuscript. I would like to congratulate on addressing an important instrument in ACT model, The Valuing Questionnaire (VQ). In my opinion, this study is relevant and is well structured, with research question well defined, relevant, and meaningful. The statistical analyses are appropriate to the aims of the study, and the results are interesting. There are some issues that I have outlined below.

Abstract
1. The abstract is well structured and presents relevant information. I just have a minor suggestion. In methods please add the following information: “The VQ was administered to a Spaniard sample of 846 adult participants from general online population”.

Introduction
2.The introduction is very complete. It was presented a good background, and objectives and hypothesis of this study was clear. I just have a question: In first paragraph of introduction, the authors use the term “middle-level term”. What this means?

Material & Methods
3. I think the description of the procedures (lines 191 to 199) could be presented before the participants (line 139).
4. What was the method used to estimate the sample size of this study?
5. What are the inclusion and exclusion criteria of this study? How many participants accepted participated in this study?

Instruments
6. Can the authors add the VQ’s alpha of the original study, as they do with the other scales?

Statistical and Psychometric Analyses
The authors presented a logical and adequate description of the statistical analyses.

Results
7. What is the correlation between two factors of the scale?

Discussion
8. I think it would be convenient to start the discussion with a sentence that would inform us about what this scale measures.
9. The authors could discuss the practical implications of the current study.

Reviewer 2 ·

Basic reporting

In my view, the manuscritp is well-written in English. The introduction section is well structured, although additional information is mainly required to strengthen the choice of this questionnaire beyond its broad use (see additional comments, section: introduction).

Experimental design

The study meets all ethical standards required for human research.
The contribution of the study, justification and research question are well defined. The objective is clear.
The type of design adopted in the study is missed. Anyway, the needed steps for validation were adequately followed.
The authors described the materials and method in an accurate and valid manner. However, some psychometric analyses are not run (see additional comments: section: method). In the same vein, the multi-group analysis of invariance is uncompleted by excluding strict invariance.
The data anlyses are in line with the objective raised.

Validity of the findings

Conclusions are consistent with the objective raised, data analyses run and results obtained. However, practical implications are needed to highlight the utility of the study (see additional comments: section: conclusions). A point underlined was to gather evidence supporting cross-cultural inviariance of the questionnaire.

Additional comments

Below, a series of comments are proposed to improve the quality of the content and methodological aspects of the manuscript.

Introduction:
Comment 1. Lines 59-60: Please, consider the inclusion of a brief definition for each one of the six core processes described in psychological flexibility.

Comment 2. Lines 87-88. The authors indicated that three systematic review studies were carried out about self-reported measures of values. However, it is unknow what they added the study. Please, more information is required on these studies.

Comment 3. Lines 90-93. The authors listed a series of measures of values to finish with the valuing questionnaire. There is a need to strengthen the choice of this questionnaire beyond its broad use. In this sense, specify the advantages or benefits that this questionnaire presents in measuring values regarding the other instruments mentioned.

Comment 4. Lines 95-105. Great work in describing the instrument to be validated.

Method
Comment 5. Design. Please, specify the type of design adopted for this study. According to Ato et al. (2013), an instrument design was followed by analysing the psychometric properties of a measurement instrument.

Comment 6. Participants. Please, add the type of method used to recruit and select the participants.

Comment 7. Line 149. This sentence “They responded to an anonymous Internet survey distributed through social media” may be removed given that similar information was found in the procedure subsection.

Comment 8. Instruments. The stem that preceded to each instrument is missed.

Comment 9. Procedure. More information on the online questionnaire’s administration process is required. To illustrate, type instructions and guidelines given to fill the online survey, strategies followed to control social desirability in completing the survey, or average time for completion.

Comment 10. Statistical and Psychometric Analysis. Please, indicate the statistical treatment followed for multivariate and univariate outliers.

Comment 11. Statistical and Psychometric Analysis: Line 216-218: Assessment of RMSEA. Please, add p-value to have additional criteria to judge the RMSEA performance (Kline, 2016, page 273-276).

Comment 12. Statistical and Psychometric Analysis: Multi-group analysis of invariance. The most commonly used methodological proposals to test invariance consistently indicate four types of invariance to be examined (e.g., Gregorich, 2006; Milfont & Fisher, 2010; Putnick & Bornstein, 2016). It is, therefore, needed to include strict invariance into the two multi-group analyses run.

Comment 13. Statistical and Psychometric Analysis: Internal consistency. It is required to include cut-off points for interpretation of Cronbach’s alpha and McDonald’s omega. Indeed, it is recommended to clarify if the interpretation of these coefficients is made on the basis of mean scores or using 95%confidence interval of this mean score.

Comment 14. Statistical and Psychometric Analysis: Discriminant validity. It is required to estimate some criterion to analyse the instrument’s discriminant validity. It is, therefore, recommended to compute heterotrait-monotrait ratio of correlations (Henseler et al., 2015). This criterion displays a good level of discrimination among factor when values are as high as .85 (Henseler et al., 2015).

Comment 15. Statistical and Psychometric Analysis: Convergent validity. It is advisable to estimate average variance extracted as a convergent validity evidence (Martínez-García & Martínez-Caro, 2009). This measure is shown to be acceptable with values equal to .50 or higher (Hair et al., 2018)

Comment 16. Statistical and Psychometric Analysis: Criterion validity: Correlations used to estimate convergent validity are more suitable to provide criterion validity evidence. In this same vein, it is suggested to consider partial correlations controlling for country, gender or age in order to gain statistical power in the type of evidence given. Moreover, the instrument’s criterion validity is missed.
Results
Comment 17. Dimensionality: Line 258. This sentence “we chose the two-factor model due to its greater parsimony” is confusing for me. Please, specify the criterion followed to affirm that the two-factor model was more parsimonious. Parsimony indices (e.g., CAIC) were not reported.

Comment 18. Table 1. Goodness-of-fit measures: If the 2/ df coefficient is estimated for both models, its value is higher than 5, cut-off point considered as minimally acceptable (Bentler, 1989). Particularly, this coefficient is 5.46 for the two-factor model, and 5.54 for the two-factor model with correlated error terms.

Comment 19. Measurement invariance: Lines 266-269 and Table 2: In my view, there is an incoherence between the sentence “changes in CFI and NNFI indexes were equal to or greater than -.01” and differences below .010 in CFI and NNFI. Please, revise this point.

Comment 20. Validity evidence based on relationships with other variables. This section has been described in the data analysis subsection as convergent validity. Please, revise this point to gain coherence and consistence.

Discussion

Comment 21. It is needed to compare the results obtained in this study with those reported in the previous validation studies on the VQ. To illustrate: lines 309-310. “Regarding internal consistency, the VQ showed appropriate Cronbach's alpha and McDonald's omega values for both Progress (.85) and Obstruction (.84)”.

Comment 22. It is required to include a discussion about the validity evidence based on relationships with other variables. In addition, it is needed to discuss the results regarding differences by gender and country.

Conclusions:
Comment 23. lines 350-352. Practical implications require to be improved.

References
Ato, M., López-García, J. J., & Benavente, A. (2013). A classification system for research designs in psychology. Annals of Psychology, 29(3), 1038–1059. https://doi.org/10.6018/analesps.29.3.178511
Bentler, P. M. (1989). EQS structural equations program manual. BMDP Statistical Software.
Gregorich, S. E. (2006). Do self-report instruments allow meaningful comparisons across diverse population groups? Testing measurement invariance using the confirmatory factor analysis framework. Medical Care, 44(11 Suppl 3), S78–S94. https://doi.org/10.1097/01.mlr.0000245454.12228.8f
Hair, J. F. J., Black, W. C., Babin, B. J., & Anderson, R. E. (2018). Multivariate data analysis (8th ed.). Cengage Learning EMEA.
Henseler, J., Ringle, C. M., & Sarstedt, M. (2015). A new criterion for assessing discriminant validity in variance-based structural equation modeling. Journal of the Academy of Marketing Science, 43(1), 115–135. https://doi.org/10.1007/s11747-014-0403-8
Kline, R. B. (2016). Principles and practice of structural equation modeling (4th ed.). The Guilford Press.
Martínez-García, J. A., & Martínez-Caro, L. (2009). Discriminant validity as a scale evaluation criterion: Theory or statistics? Universitas Psychologica, 8(1), 27–36.
Milfont, T. L., & Fisher, R. (2010). Testing measurement invariance across groups: Applications in cross-cultural research. International Journal of Psychological Research, 3(1), 111–121. https://doi.org/10.21500/20112084.857
Putnick, D. L., & Bornstein, M. H. (2016). Measurement invariance conventions and reporting: The state of the art and future directions for psychological research. Developmental Review, 41, 71–90. https://doi.org/10.1016/j.dr.2016.06.004

---

## Round 0.2 · accepted · Accept

Authors, the reviewers have checked your revised manuscript, and are satisfied with the improved work. The editor is also satisfied and considers the current form acceptable for publication. Thank you for finding PeerJ as your journal of choice and look forward to your future scholarly contributions.
Congratulations and very best regards.

Reviewer 1 ·

Basic reporting

The manuscript was improved. Congratulations! I have a minor suggestion in the introduction:
on the last page of the introduction, I suggest you replace the word "Unfortunately" to "However". We need to pay attention to the words used in scientific manuscripts.

Experimental design

How did the researches ensure that participants who answered the questionnaire belonging to Sample 1 did not also complete the questionnaire belonging to Sample 2?

Validity of the findings

no comment

Additional comments

The manuscript was improved. Congratulations!

Reviewer 2 ·

Basic reporting

The authors made a great labour in responding to every comment raised by me. The manuscript is well structured. The introduction have a goold guiding thread and the study contribution is clear. The objectives are clear.

Experimental design

The method section is well structured and contained all information necessary to replicate this study. Indeed, the data analyses is aligned with the proposed objective and the aurhos have complemented construct validity evidence by considering the suggested coefficents for discriminant and convergent validity. Similarly, the study desing is also clear owing to the authors followed the desing proposed.

Validity of the findings

The results are clear and provided an answer to the objective. The discussion is well structured and suitably argued the results obatined wiht the ones in previous studies. The conclusions relied on the found results and provided a response to the objective of the study.

Additional comments

No additional comments.